# Brief communication: Rapid machine learning-based extraction and measurement of ice wedge polygons in high-resolution digital elevation models

Charles J. Abolt[1,2], Michael H. Young[2], Adam L. Atchley[3], Cathy J. Wilson[3]

[1]Department of Geological Sciences, The University of Texas at Austin, Austin, TX, USA
[2]Bureau of Economic Geology, The University of Texas at Austin, Austin, TX USA
[3]Earth and Environmental Sciences Division, Los Alamos National Laboratory, Los Alamos, NM, USA

*Correspondence to*: Charles J. Abolt (chuck.abolt@beg.utexas.edu)

**Abstract.** We present a workflow for rapid delineation and microtopographic characterization of ice wedge polygons within high-resolution digital elevation models. At the core of the workflow is a convolutional neural network used to detect pixels representing polygon boundaries. A watershed transformation is subsequently used to segment imagery into discrete polygons. Fast training times (<5 minutes) permit an iterative approach to improving skill as the routine is applied across broad landscapes. Results from study sites near Barrow and Prudhoe Bay, Alaska demonstrate robust performance in diverse tundra settings, with manual validations demonstrating 70-96% accuracy by area at the kilometer scale. The methodology permits precise, spatially extensive measurements of polygonal microtopography and trough network geometry.

## 1 Introduction and Background

This research addresses the problem of delineating and measuring ice wedge polygons within high-resolution digital elevation models (DEMs). Ice wedge polygons are the surface expression of ice wedges, a form of ground ice nearly ubiquitous to coastal tundra environments in North America and Eurasia (Leffingwell, 1915; Lachenbruch, 1962). High resolution inventories of ice wedge polygon microtopography are of hydrologic and ecologic interest, because decimeter-scale variability in polygonal relief can drive pronounced changes to soil drainage (Liljedahl et al., 2016), and surface emissions of $CO_2$ and $CH_4$ (Lara et al., 2015; Wainwright et al., 2015). At typical sizes, several thousand ice wedge polygons may occupy a single square kilometer of terrain, motivating our development of an automated approach to mapping. The key innovation in our method is the use of a convolutional neural network (CNN), a variety of machine learning algorithm, to identify pixels representing polygon boundaries. Integrated within a set of common image processing operations, this approach permits the extraction of microtopographic attributes from entire populations of ice wedge polygons at the kilometer scale or greater.

Previous geospatial surveys of polygonal microtopography have often aimed to map the occurrence of two geomorphic endmembers: basin-shaped low-centered polygons (LCPs), which are characterized by rims of soil at the perimeters, and hummock-shaped high-centered polygons (HCPs), which often are associated with permafrost degradation.

Analyses of historic aerial photography have demonstrated a pan-Arctic acceleration since 1989 in rates of LCP conversion into HCPs, a process which improves soil drainage and stimulates enhanced emissions of $CO_2$ (Jorgenson et al., 2006; Raynolds et al., 2014; Jorgenson et al., 2015; Liljedahl et al., 2016). Nonetheless, precise rates of geomorphic change have been difficult to quantify, as these surveys typically have relied on proxy indicators, such as the presence of ponded water in deepening HCP

troughs. In a related effort to characterize contemporary polygon microtopography, a landcover map of LCP and HCP occurrence across the Arctic coastal plain of northern Alaska was recently developed using multispectral imagery from the Landsat 8 satellite at 30 m resolution (Lara et al., 2018). This dataset offers a static estimate of variation in polygonal form over unprecedented spatial scales; however, geomorphology was inferred from the characteristics of pixels larger than typical polygons, preventing inspection of individual features.

Higher-resolution approaches to segment imagery into discrete ice wedge polygons have often been motivated by efforts to analyse trough network geometry. On both Earth and Mars, for example, paleo-environmental conditions in remnant polygonal landscapes have been inferred by comparing parameters such as boundary spacing and orientation with systems in modern periglacial terrain (*e.g.*, Pina et al., 2008; Levy et al., 2009; Ulrich et al., 2011). An early semi-automated approach to delineating Martian polygons from satellite imagery was developed by Pina et al. (2006), who employed morphological image

processing operations to emphasize polygonal boundaries, then applied a watershed transformation (discussed in Section 4.1.3) to identify discrete polygons. This workflow was later applied to lidar-derived DEMs from a landscape outside Barrow, Alaska by Wainwright et al. (2015), but in their experience and our own, robust results at spatial scales approaching a square kilometer or greater were elusive.

Our application of CNNs to the task of identifying polygonal troughs was inspired by the remarkable solutions that

CNNs recently have permitted to previously intractable image processing problems. Aided by advances in the performance of graphics processing units (GPUs) over the last decade, CNNs have demonstrated unprecedented skill at tasks analogous to ice wedge polygon delineation, such as cell membrane identification in biomedical images (Ciresan et al., 2012) or road extraction from satellite imagery (Kestur et al., 2018; Xu et al., 2018). Motivated by this potential, an exploratory study was recently conducted by Zhang et al. (2018), who demonstrated that a sophisticated neural network, the Mask R-CNN of He et al. (2017),

is capable of end-to-end extraction of ice wedge polygons from satellite-based optical imagery, capturing ~79% of ice wedge polygons across a >134 km$^2$ field site and classifying each as HCP or LCP. The authors concluded that the method has potential for pan-Arctic mapping of polygonal landscapes. Here we explore an alternative approach, using a less complex CNN paired with a set of post-processing operations, to extract ice wedge polygons from high-resolution DEMs derived from airborne Lidar surveys. An advantage to this method is that training the CNN is rapid (~5 minutes or less on a personal laptop),

permitting an iterative workflow in which supplementary data can easily be incorporated to boost skill in targeted areas. We demonstrate the suitability of this approach to extract ice wedge polygons with very high accuracy (up to 96% at the kilometer scale), applying it to ten field sites of 1 km$^2$ outside Barrow and Prudhoe Bay, Alaska. Because our method operates on high resolution elevation data, it enables direct measurement of polygonal microtopography, and we anticipate that in the future, the method will permit precise monitoring of surface deformation in landscapes covered by repeat airborne surveys.

## 2 Study areas and data acquisition

To demonstrate the flexibility of our approach, we applied it simultaneously at two clusters of study sites near Barrow and Prudhoe Bay, Alaska, settings with highly divergent ice wedge polygon geomorphology, ~300 km distant from one another (Fig S1).

### 2.1 Barrow

The first cluster of study sites (Figs. S2-S3) is located within 10 km of the Beaufort Sea coast in the Barrow Environmental Observatory, operated by the National Environmental Observatory Network (NEON). Mean elevation is less than 5 m above sea level, and vegetation consists of uniformly low-growing grasses and sedges. Mesoscale topography is mostly flat but marked by depressions up to 2 m deep associated with draws and drained lake beds. In the landcover map of Lara et al. (2018), the area is characterized by extensive coverage by both LCPs and HCPs, with occasional lakes and patches of non-polygonal meadow. Microtopography at the sites reflects nearly ubiquitous ice wedge development, which becomes occluded in some of the depressions. Ice wedge polygons are of complex geometry and highly variable area, ranging from ~10 $m^2$ to >2000 $m^2$. An airborne lidar survey was flown in August 2012 as part of the U.S. Department of Energy's Next Generation Ecosystems Experiment-Arctic program (https://ngee-arctic.ornl.gov/). The resulting point cloud was processed into a 25 cm horizontal resolution DEM with an estimated vertical accuracy of 0.145 m (Wilson et al., 2012). In the present study, to compare algorithm performance on data of variable spatial resolution, the 25 cm DEM was resampled at 50 cm and 100 cm resolution. Two sites of 1 $km^2$, here referred to as Barrow-1 and Barrow-2, were extracted from the DEMs and processed using our workflow.

### 2.2 Prudhoe Bay

The second cluster of sites (Figs. S4-S11) is approximately 300 km east of the first and farther inland, located ~40 km south of Prudhoe Bay, AK (Fig S1). As at Barrow, vegetation consists almost exclusively of low and even-growing grasses and sedges. Mesoscale topography is generally flat, with a slight (<4%) dip toward the northwest. In the landcover map of Lara et al. (2018), the area is primarily characterized by HCPs, with smaller clusters of LCPs, patches of non-polygonal meadow, and occasional lakes. Ice wedge polygons are generally of more consistent area than those of Barrow, ~400-800 $m^2$. Airborne lidar data was acquired in August 2012 by the Bureau of Economic Geology at the University of Texas at Austin (Paine et al., 2015) and subsequently processed into 25 cm, 50 cm, and 100 cm resolution DEMs. Vertical accuracy was estimated at 0.10 m. As the Prudhoe Bay survey area is substantially larger than the survey area at Barrow, eight sites of 1 $km^2$, here referred to as Prudhoe-1 through Prudhoe-8, were extracted from the DEMs and processed using our workflow.

## 3 Methods

### 3.1 Polygon delineation algorithm

A chart summarizing our iterative workflow is presented in Fig. 1, and several intermediate stages in the polygon delineation algorithm are illustrated in Fig. 2. In the first (pre-processing) stage, regional trends were removed from a DEM (Fig. S12), generating an image of polygonal microtopography (Fig. 2A). Next, the microtopographic information was processed by a CNN, which was trained to use the 27×27 neighborhood surrounding each pixel to assign a label of "boundary" or "not boundary" (Fig. 2B). A distance transformation was then applied, (*i.e.*, each non-boundary pixel was assigned a negative intensity proportional to its Euclidean distance from the closest boundary), generating a grayscale image analogous to a DEM of isolated basins, in which the polygonal boundaries appear as ridges (Fig. 2C). Subsequently, a watershed transform was applied to segment the image into discrete ice wedge polygons (Fig. 2D). These steps, and a post-processing algorithm used to remove non-polygonal terrain from the final image, are described in detail below.

### 3.1.1 Pre-processing

In the pre-processing stage, regional topographic trends were estimated by processing the DEM with a 2D filter, which assigned to each pixel the mean elevation within a 20 m radius. This radius was chosen such that the area over which elevation was averaged would be larger than a typical ice wedge polygon. Polygon-scale microtopography was then estimated by subtracting the regional topography from the DEM (Fig S12). In preparation for passing the data to the CNN, microtopography was subsequently converted to 8-bit gray-scale imagery. The minimum intensity (0) was assigned to depressions of 0.7 m or greater, and the maximum intensity (255) was assigned to ridges of 0.7 m or greater. These bounds captured >99% of pixel values at each study site. Finally, one thumbnail-sized image was created for each pixel in the microtopography raster, capturing the immediate neighbourhood surrounding it. These thumbnail images were the direct input to the CNN. The CNN required the width in pixels of each thumbnail to be an odd multiple of 9; therefore, at 50 cm resolution the thumbnails were assigned a width of 27 pixels (13.5 m), at 25 cm resolution a width of 45 pixels (11.25 m), and at 100 cm resolution a width of 27 pixels (27 m). The width of these thumbnails was chosen such that each image would contain sufficient spatial context for a human observer to distinguish easily between polygonal boundaries, which typically were demarcated by inter-polygonal troughs, and other microtopographic depressions such as LCP centers.

### 3.1.2 Convolutional neural network

The function of the CNN in our workflow was to identify pixels likely to represent boundaries. Conceptually, a CNN is a classification tool that accepts images of a fixed size (in our case, the thumbnails described in the previous section) as input and generates categorical labels as output. The CNN determines decision criteria through training with a set of manually-labeled images. The architecture of a CNN consists of a user-defined sequence of components, or layers, which take inspiration from the neural connections of the visual cortex. We developed our CNN in MATLAB (R2017b) using the Image Processing,

Parallel Computing, and Neural Network toolboxes. We purposefully constructed the CNN with an architecture of minimal complexity, to maximize the efficiency of training and application. Here we briefly describe the function of each layer in our CNN; for more detailed description, the reader is directed to Ciresan et al. (2012).

Summarized in Table S1, the most important components of our CNN were a single convolutional layer, a max-pooling layer, and two fully connected layers. In the convolutional layer, a set of 2D filters was applied to the input image, generating intermediate images in which features including concavities, convexities, or linear edges were detected. The max-pooling layer downsized the height and width of these intermediate images by a factor of three, by selecting the highest intensity pixel in a moving 3×3 window with a stride of 3 pixels. Each pixel in the downsized intermediate images was then passed as an input signal to the fully connected layers, which functioned identically to standard neural networks. Two additional components of our CNN were Rectified Linear Unit (ReLU) layers, which enhance non-linearity by reassigning a value of zero to any negative signals output by a preceding layer, and a softmax layer, which converted the output from the final fully connected layer into a probability for each categorical label (*i.e.*, boundary or not boundary).

During training, the weights of the 2D filters in the convolutional layer and the activation functions of the neurons in the fully connected layers were optimized to correctly predict the labels in a training deck of images. Our workflow was designed to generate the training deck primarily by processing 100 ×100 m tiles of manually-labeled imageryIn each of these tiles, boundary pixels were delineated by hand in a standard raster graphics editor, a process that required ~1 hour per tile at 50 cm resolution (Fig. S13A). Our algorithm imported these tiles, identified the geographic coordinates of each pixel identified as a boundary, then created a thumbnail image centered on that pixel from the 8-bit microtopographic imagery. This procedure generated several thousand thumbnail images centered on boundaries from each manually delineated tile. Subsequently, an equal number of pixels not labeled as boundaries were selected at random, and the thumbnail extraction procedure was repeated, generating a set of non-boundary images for the training deck. Finally, for more targeted training that did not require full delineation of a 100 ×100 m tile, individual instances of boundary or non-boundary features could also be added to supplement the training deck, based again on manual delineation (Fig. S13B). Just prior to training, 25% of the training deck assembled by these methods was set aside to be used for validation.

Once trained, the CNN was executed to assign a label of "boundary" or "not boundary" to the thumbnail image corresponding to each pixel of a study site. These labels were then reassembled into a binary image of polygon boundaries (Fig. 2B), which was further processed to extract discrete ice wedge polygons.

### 3.1.3 Polygon extraction

After applying the CNN to classify all pixels at a site as boundary or not boundary, we extracted discrete ice wedge polygons by applying several standard image processing operations. The first step was elimination of "salt and pepper" noise in the binary image, which we accomplished by eliminating all contiguous sets of boundary-identified pixels with an area $< 20$ m$^2$. This threshold was selected based on the reasoning that most true boundary pixels should be part of a continuous network, covering an area arbitrarily larger than 20 m$^2$, while most false detections should occur in smaller clusters. Next, we

applied a distance transform, assigning to every non-boundary pixel a negative intensity equal to its Euclidean distance from the nearest boundary. This created an intermediate image in which each ice wedge polygon appeared as a valley, surrounded on all or most sides by ridges representing the ice wedge network (Fig. 2C). At this stage, to prevent over-segmentation, valleys with maximum depths of 1.5 meters or less were then identified and merged with the closest neighbor through morphological reconstruction (Soille, 1992). The effect of this procedure was to ensure that the algorithm would only delineate polygons whose centers contained at least one point greater than 1.5 m from the boundaries, as field observations indicate that ice wedge polygons tend to measure at least several meters across (Leffingwell, 1915; Lachenbruch, 1962). Next, watershed segmentation was applied to divide the valleys into discrete polygons (Fig. 2D). Our use of this operation was inspired by its incorporation in the polygon delineation method developed by Pina et al. (2006). Conceptually, this procedure was analogous to identifying the up-gradient region or area of attraction surrounding each local minimum.

### 3.1.4 Partitioning of non-polygonal ground

In the final stage of delineation, we partitioned out regions of a survey area that had been segmented using the techniques described above, but were unlikely to represent true ice wedge polygons. For example, polygons were eliminated from the draw in the southern half of Barrow-1 (Fig. S2A), where microtopography was too occluded to permit accurate delineation. Toward this aim, our algorithm analysed individually each boundary between two polygons (black line segments in Fig. 2D), tabulating the number of pixels that had been identified positively by the CNN (white pixels in Fig. 2B). It then dissolved all boundaries in which less than half the pixels had been classified positively, merging adjacent polygons. In practice, this procedure resulted in areas of non-polygonal terrain being demarcated by unusually large "polygons." We removed these areas by partitioning out any polygon with an area greater than 10,000 $m^2$, a threshold selected to be arbitrarily larger than most real ice wedge polygons. This procedure had the strengths of being conceptually simple and providing a deterministic means of partitioning non-polygonal terrain from the rest of a survey area.

### 3.2 Microtopographic analysis

To demonstrate the capabilities of our workflow for microtopographic analysis, we developed a simple method for measuring the relative elevation at the center of each delineated polygon, serving as a proxy for LCP or HCP form. In each polygon, we first applied a distance transform, calculating the distance from the closest boundary of all interior pixels. We then divided the area of the polygon in half at the median distance from boundaries, designating a ring of "outer" pixels and an equally sized core of center pixels. Microtopographic relief was then estimated as the difference in mean elevation between the center and outer pixels.

### 3.3 Case study experimental design

The case study was first conducted using topographic data at 50 cm resolution, then repeated at 25 cm and 100 cm resolution. Training was focused primarily on sites Prudhoe-1 and Barrow-1. Leveraging the rapid training and application

times of our CNN, we manually delineated one $100 \times 100$ m tile of imagery at a time from either site, trained the CNN, extracted results from both sites, then introduced additional training data from regions of poor performance to improve skill (Fig. 1). After four iterations of this approach, the CNN incorporated training data from three fully-delineated tiles at Barrow-1 and one at Prudhoe-1, representing 3% and 1% of the sites, respectively. From this point, we opted to "fine-tune" the CNN by

supplementing the training deck directly with instances of problematic features, rather than using information from fully-delineated tiles. Several examples of boundary and non-boundary features were included from Barrow-1 and Prudhoe-1. Next, to test its extensibility, the CNN was applied across the remaining sites, and re-trained once more. In this final iteration, several instances of boundary and non-boundary features (but no fully delineated tiles) were incorporated into the training deck from sites Prudhoe-2, Prudhoe-3, and Prudhoe-4. No training data at all were incorporated from sites Prudhoe-5 through Prudhoe-8

or Barrow-2. (All training data used in the final iteration of our workflow can be viewed in the data and code repository accompanying this article.) Once this procedure was complete at 50 cm resolution, training decks at 25 cm and 100 cm resolution were prepared. To generate CNNs comparable to the network trained on 50 cm data, the 25 cm and 100 cm training decks were constructed using data sampled from identical geographic locations, but manual labeling was performed without reference to the labeled 50 cm resolution data.

After the CNN was trained and applied across all study sites, we quantified the performance of the polygon delineation algorithm through manual validation. At each site, we first calculated the total area and number of polygons extracted from the landscape. We then randomly sampled 500 of the computer-delineated polygons, and classified each as either whole, fragmentary, conglomerate, or false. Fragmentary polygons were defined as computer-delineated polygons that included less than 90% of one real polygon; conglomerate polygons were defined as computer-delineated polygons comprising parts of two

or more real polygons; and false polygons were defined as computer-delineated polygons occupying terrain in which no polygonal pattern was deemed visible to the human evaluator. The percentage of computer-delineated polygonal terrain corresponding to each class was then calculated by number of polygons and by area. This procedure was completed for all ten study sites at 50 cm data resolution, and for sites Prudhoe-1 and Barrow-1 at 25 cm and 100 cm resolution.

## 4 Results and discussion

### 4.1 Training speed and accuracy

Due to the compact architecture of our CNN, training speeds at 50 cm resolution and 100 cm resolution were rapid. At 50 cm resolution, the training procedure operated on a deck of ~36,000 thumbnail images. Executed on a personal laptop with an Intel i7 CPU and a single GeForce MX150 GPU, accuracies >97% on the training deck and >95% on the validation deck of thumbnails were achieved in less than five minutes. At 100 cm resolution, the procedure operated on ~12,000 thumbnail

images, achieving comparable levels of accuracy within 90 seconds. These speeds enable the iterative approach to training on which our workflow is based (Fig. 1), as the CNN can be re-trained quickly to incorporate new data when applied across increasingly large areas.

Using 25 cm resolution data, the training procedure operated on a set of ~115,000 thumbnail images. Accuracies >97% on the training deck and >95% on the validation deck were once more obtained, but training required just under one hour on the same computer. This substantial increase to training time is attributable to the facts that more thumbnail images were processed, and the number of pixels in each thumbnail was larger, making execution of the CNN more computationally expensive.

## 4.2 Delineation speed and validation

Operating at 50 cm resolution, delineation of ice wedge polygons within a 1 km$^2$ field site required ~2 minutes, including application of the CNN and subsequent post-processing. Results generally were very accurate; across study sites, ~1000-5000 ice wedge polygons were detected per square kilometer, of which 85-96% were estimated as "whole" during manual validation, representing 70-96% of the polygonal ground by area (Table 1). The most common type of error at all sites with <95% accuracy was incorrect aggregation of several real ice wedge polygons into a single feature. Unsurprisingly, performance was strongest at sites with clearly defined polygon boundaries and relatively simple polygonal geometry, such as Prudhoe-1, Prudhoe-7, and Prudhoe-8 (Figs. S4, S10, S11). In contrast, performance was weakest at sites such as Barrow-2 or Prudhoe-6 (Figs. S2, S9), where considerable swaths of terrain are characterized by faint microtopography, as ice wedge polygons appear to grade into non-polygonal terrain. In such locations, polygonal boundaries frequently went undetected, resulting in the delineation of unrealistically large conglomerate polygons. In general, the results of the delineation clearly illustrate that the polygonal network at Barrow possesses more complex geometry than Prudhoe Bay, with many instances where secondary or tertiary ice wedges appear to subdivide larger ice wedge polygons.

Although simple, our post-processing procedure for partitioning out non-polygonal ground from the results was generally accurate. Examples of features successfully removed from the 50 cm resolution DEMs included thaw lakes (Figs. S6-S10), drained thaw lake basins (Fig. S2), stream beds (Figs. S5, S9), non-polygonal marsh (Figs. S3, S7, S9), shallow ponds (Fig. S7), and the flood plain of a braided stream (Fig. S10). Because the partitioning procedure removed areas with a low density of boundaries identified by the CNN, we found that it could be trained efficiently by supplementing the training deck with extra examples of "non-boundary" thumbnails extracted from these regions. Across the study sites, we encountered almost no cases where well-defined polygons were mistakenly partitioned out. Instances of non-polygonal ground mistakenly classified as polygonal accounted for less than 1% of machine-delineated polygons by area.

When the delineation algorithm was repeated using data at 100 cm resolution, delineation speeds increased somewhat, but performance dropped significantly, with greater declines occurring in the challenging environment of Barrow (Table S2). We attribute declines in performance to an obscuration of fainter polygonal boundaries at this resolution, and to decreases in the amount of contextual information that can be derived from the neighbourhood of any given pixel, reducing the capacity for the algorithm to distinguish between polygonal troughs and other microtopographic depressions (Fig. S14). Somewhat surprisingly, performance also declined slightly using data at 25 cm resolution, with larger fractions of fragmentary and false polygons accounting for most of the increases in errors (Table S2). We attribute these mistakes to the larger number of

distinguishable features in the higher resolution data, leading the algorithm to more frequently mistake features not encountered in the training dataset as boundary pixels. As the imagery nonetheless is of sufficient resolution for our purposes, we anticipate that, with augmentations to the training dataset, performance at 25 cm resolution could improve and even exceed performance at 50 cm resolution; however, performance speeds were generally inhibitory to our workflow, as delineation required ~50

minutes per square kilometer. We therefore conclude that 50 cm resolution data is optimal for our analysis.

### 4.3 Measurement of polygonal microtopography

The time required to execute our procedure for measuring polygonal microtopography varied from ~10-30 seconds per square kilometer at 50 cm resolution, depending on the number of polygons delineated. A comparison of calculated relief at polygon centers between Prudhoe-1 and Barrow-1 reveals that both sites are characterized by the prevalence of HCPs, which

surround smaller clusters of LCPs (Fig. 3). Relief tends to be more extreme at Barrow, with the relative elevation of polygon centers commonly approaching 20-30 cm. Our automated calculations of relief align well with visual inspection of the DEMs, as rimmed LCPs are consistently assigned negative center elevations. To our knowledge, our results represent the first direct measurement of polygonal relief at the kilometer-scale, demonstrating a spectrum of center elevations rather than a binary classification into LCP or HCP. We anticipate these measurements may be useful for further investigations into relationships

between microtopography, soil moisture, and carbon fluxes (*e.g.*, Wainwright et al., 2015).

### 4.4 Comparison with Mask R-CNN and future applications

Several key differences are apparent between the workflow presented in this paper (hereafter termed the CNN-watershed approach) and a recent implementation of Mask R-CNN for mapping of ice wedge polygons (Zhang et al., 2018), revealing relative strengths to each approach. An advantage of the CNN-watershed approach, stemming from its sparse neural

architecture, is that training times are extremely rapid, facilitating iterative improvements to skill (Fig. 1). In contrast, although inference times using the CNN-watershed approach are reasonable, extraction of ice wedge polygons over broad landscapes is several times faster using Mask R-CNN, with a reported time of ~21 minutes for inference over 134 km$^2$ of terrain (Zhang et al., 2018). Because the CNN-watershed approach operates on high-resolution DEMs, it enables direct quantification of polygonal relief, whereas Mask R-CNN instead produces a binary classification of each polygon as either LCP or HCP. The

CNN-watershed approach is therefore useful for generating unique datasets summarizing polygonal geomorphology, demonstrating high-performance at spatial scales typical of airborne surveys using lidar or photogrammetry to produce high-resolution DEMs. In comparison, because Mask R-CNN has been trained to operate on satellite-derived optical imagery with global coverage, it is uniquely well-suited for application across very broad regions, with potential to generate pan-Arctic maps of land cover by polygon type. Because of differences in training and inference procedures and the spatial scales at which they

ideally operate, the training data requirements and accuracy of the two approaches are difficult to compare directly; nonetheless, in several aspects, performance appears to be similar (Text S1).

Because the CNN-watershed approach generates direct measurements of polygonal microtopography, one application to which it is uniquely amenable is precise monitoring of microtopopographic deformation in areas covered by repeat airborne surveys. Through such analysis, we anticipate that it will permit polygon-level quantification of ground subsidence over timespans of years, potentially yielding new insights into the vulnerability of various landscape units to thermokarst. An additional research problem to which the CNN-watershed approach is well-suited is quantification of polygonal network parameters, such as boundary spacing and orientation, to explore relationships to environmental factors such as climate (*e.g.*, Pina et al., 2008; Ulrich et al., 2011). These boundaries (black line segments in Figs 2D, 3B, 3D) are naturally delineated through implementation of the watershed transformation. We acknowledge that, because the surface expression of ice wedges is sometimes subtle or non-existent, ground-based delineation methods are the highest-accuracy approach to mapping ice wedge networks (Lousada et al., 2018). Nonetheless, by segmenting machine-delineated networks into individual boundaries, our workflow permits the estimation of network statistics at spatial scales unattainable through on-site surveying.

## 5 Conclusions

A relatively simple CNN paired with a set of common image processing techniques is capable of extracting polygons of highly variable size and geometry from high-resolution DEMs of diverse tundra landscapes. Successful application of the CNN is facilitated by its sparse neural architecture, which permits rapid training, testing, and incorporation of new data to improve skill. The optimal spatial resolution for DEMs processed using the workflow is ~50 cm. Potential applications for the technology include: generation of high-resolution maps of land cover by polygon type, precise quantification of microtopographic deformation in areas covered by repeat airborne surveys, and rapid extraction of center elevations and boundary parameters including spacing and orientation. These capabilities can improve understanding of environmental influences on network geometry and facilitate assessments of contemporary landscape evolution in the Arctic.

**Acknowledgements**

We are grateful for the support provided for this research, which included: the Next Generation Ecosystem Experiments Arctic (NGEE-Arctic) project (DOE ERKP757) funded by the Office of Biological and Environmental Research in the U.S. Deparment of Energy Office of Science, and the NASA Earth and Space Science Fellowship program, for an award to CJA (80NSSC17K0376). We thank Ingmar Nitze (Alfred Wegener Institute for Polar and Marine Research) and Weixing Zhang (University of Connecticut) for highly constructive feedback during peer review, and Dylan Harp (Los Alamos National Laboratory) for helpful conversations during manuscript preparation. We acknowledge the Texas Advanced Computing Center (TACC) at The University of Texas at Austin for providing HPC resources that have contributed to the results reported within this paper.

**Data availability**

Data and code are available at https://doi.org/10.5821/zenodo.2537167.

**Competing interests**

The authors declare they have no conflict of interest.

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

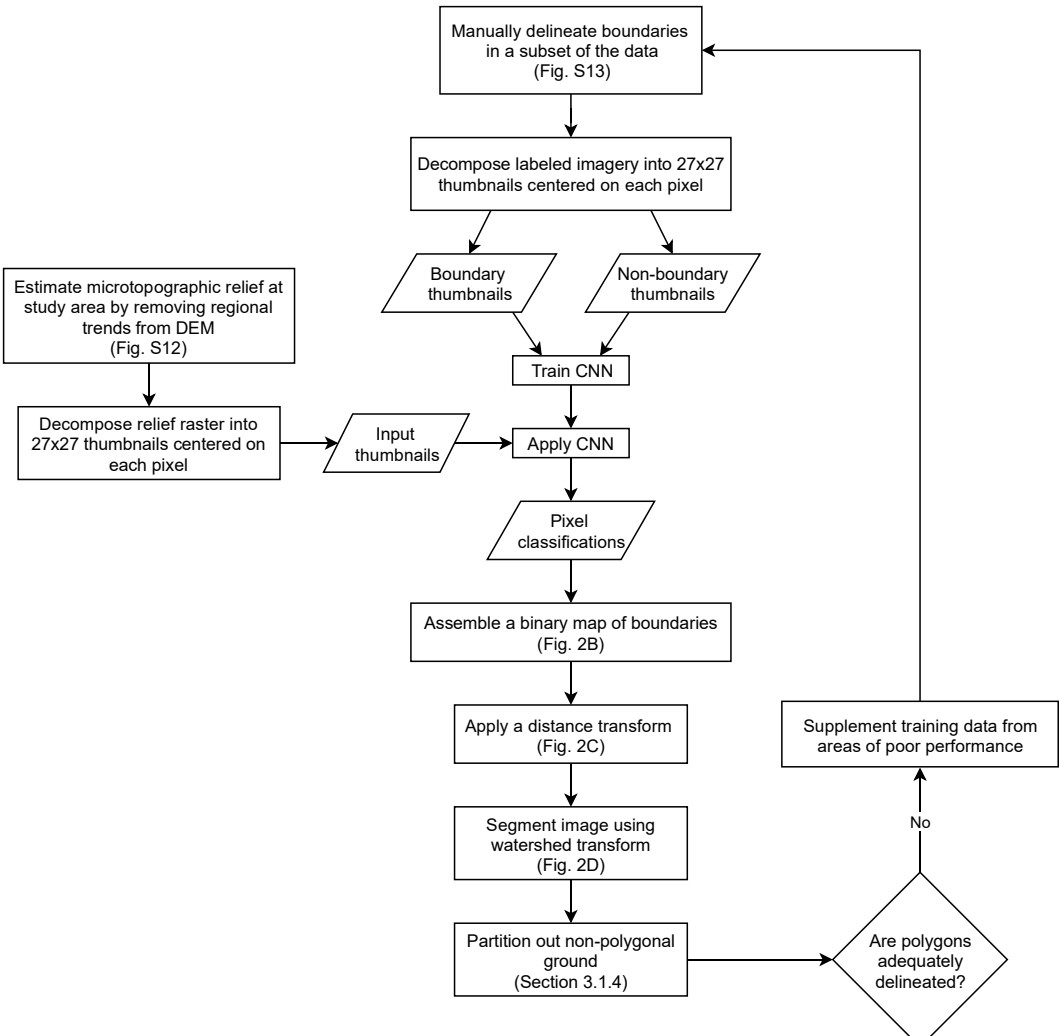

**Figure 1.**     Schematic of our iterative workflow for polygon delineation.

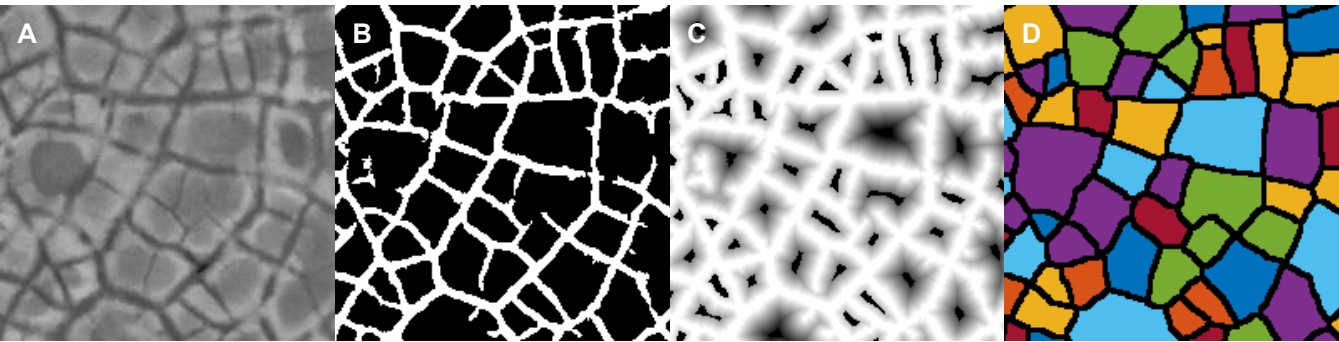

**Figure 2.**    Illustration of several intermediate stages in our workflow. The CNN processes information stored in 8-bit grayscale imagery representing microtopography (A), estimated by removing regional trends from the lidar DEM (Fig S2). The CNN identifies pixels likely to represent troughs (B). Each non-trough pixel is assigned a negative intensity equal to its distance from a trough (C) and a watershed transformation is applied to segment the image into discrete polygons (D) (colors randomly applied to emphasize polygonal boundaries).

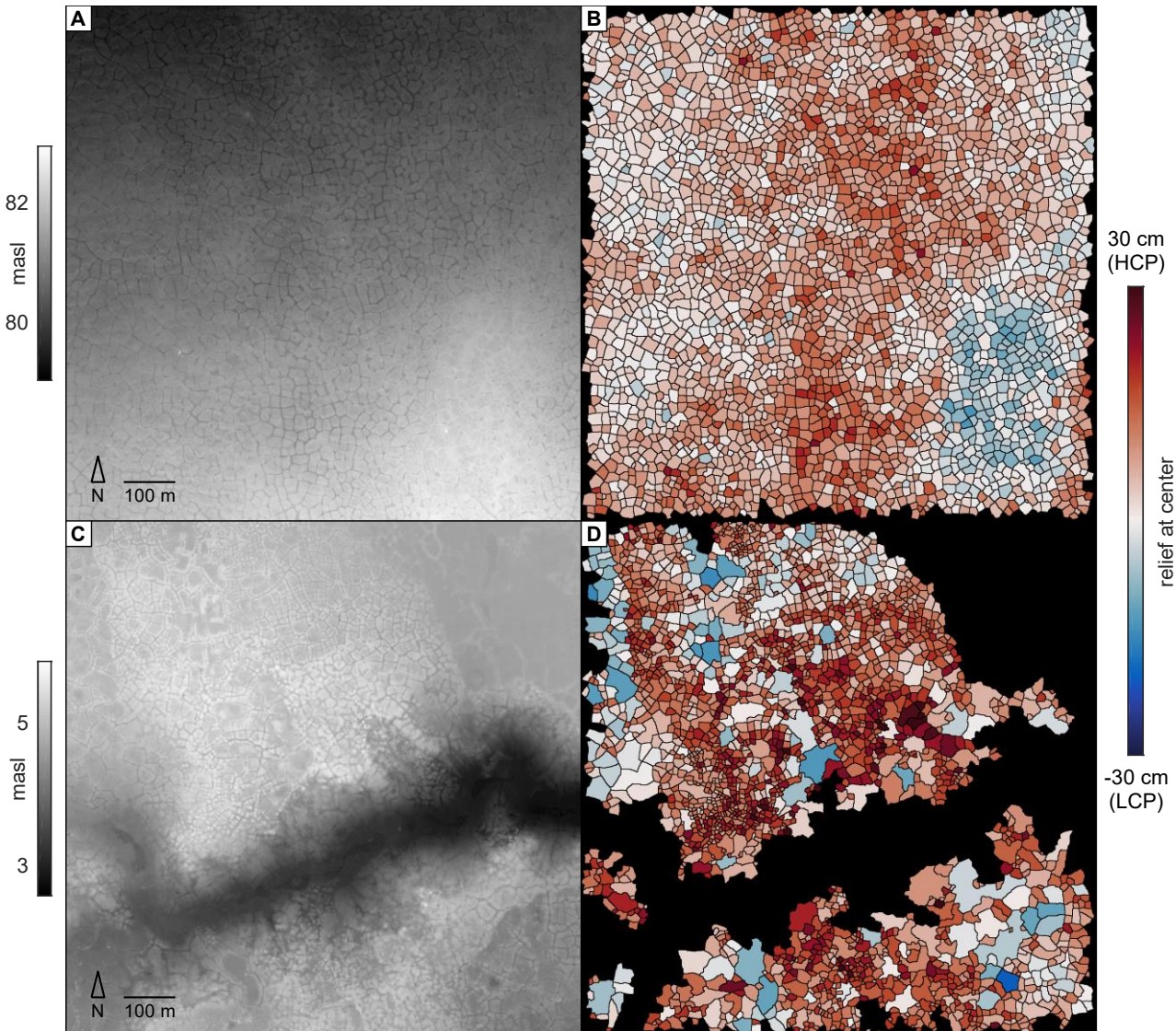

**Figure 3.** DEMs and estimates of polygonal relief at sites Prudhoe-1 (A, B) and Barrow-1 (C, D). Polygons crossing the boundaries of study sites are removed in (B, D).

**Table 1.**         Results of manual validation at 50 cm data resolution (sites are 1 km$^2$).

| Site | Polygons identified | Polygonal area (%) | % of polygons by instance | | | | % of polygons by area | | | |
|------|------|------|------|------|------|------|------|------|------|------|
| | | | Whole | Fractional | Conglomerate | Non-polygonal | Whole | Fractional | Conglomerate | Non-polygonal |
| Barrow-1 | 2555 | 68.3 | 94.2 | 2.2 | 2.2 | 1.4 | 86.9 | 1.4 | 7.5 | 4.2 |
| Barrow-2 | 2613 | 68.8 | 91.6 | 2.8 | 5.6 | 0.0 | 79.9 | 3.6 | 16.5 | 0.0 |
| Prudhoe-1 | 3227 | 99.9 | 95.6 | 2.8 | 1.4 | 0.2 | 96.3 | 2.2 | 1.4 | 0.0 |
| Prudhoe-2 | 4685 | 94.5 | 91.6 | 1.4 | 7.0 | 0.0 | 87.7 | 1.7 | 10.6 | 0.0 |
| Prudhoe-3 | 1112 | 48.2 | 91.2 | 3.6 | 5.2 | 0.0 | 82.3 | 4.2 | 13.4 | 0.0 |
| Prudhoe-4 | 1956 | 60.4 | 88.2 | 4.2 | 7.2 | 0.4 | 81.4 | 3.2 | 14.9 | 0.5 |
| Prudhoe-5 | 2969 | 77.9 | 91.8 | 4.0 | 4.2 | 0 | 87.7 | 1.7 | 10.6 | 0 |
| Prudhoe-6 | 1605 | 65.5 | 85.8 | 5.4 | 8.2 | 0.4 | 69.5 | 4.8 | 22.1 | 3.4 |
| Prudhoe-7 | 1348 | 47.3 | 94.0 | 2.2 | 3.6 | 0.2 | 90.5 | 1.7 | 7.9 | 0.0 |
| Prudhoe-8 | 3288 | 100 | 96.0 | 2.2 | 1.6 | 0.2 | 95.2 | 1.5 | 3.2 | 0.0 |