# Peer review of "Brief communication: Rapid machine learning-based extraction and measurement of ice wedge polygons in high-resolution digital elevation models"

_The Cryosphere, 2018_

## Referee Comment (RC1) · I. Nitze (Referee) · 21 Sep 2018

General summary: The manuscript with the title "Rapid machine learning-based extraction and measurement of ice wedge polygons in airborne lidar data" describes the application of novel convolutional neural network (CNN) image recognition concepts for the delineation of ice-wedge polygons from DEM data. The methodology was tested in two different sites in northern Alaska. The application of state-of-the-art image recognition methods is a rather new and unexplored approach in remote sensing applications of the cryosphere. The paper has a strong technical focus and describes the methods thoroughly. Delineating ice-wedge polygon networks is an essential task for quantify-

ing ground ice, predicting the resilience against degradation and understanding local scale hydrology. In contrast to the positive novelty of the paper, this paper lacks several major points. The authors remain very vague in the results sections, with practically no quantification or accuracy assessment of the results. To the reader it is hard to estimate the accuracy strengths and weaknesses of the method, as the results are presented rather qualitatively. Furthermore, the title suggests that the authors used Lidar data as their key input. This is somewhat misleading, as they used DEMs, which are based on Lidar data, but could be technically processed from other sources. As this manuscript has a technical focus I would really like to see a flowchart in this paper, as this will help to follow the processing chain much better. Furthermore, the authors did not mention any software (programming languages, packages) they used, which might be interesting for the readers. For the review process I am interested to see the code and the data. Overall I see a good potential for publication due to the interesting application of novel image recognition methods for delineating IW-polygons. However, the manuscript needs improvement in several sections, particularly in the results and discussion section. Therefore I recommend a major revision. Specific comments are stated below.

1. Title: The title is somewhat misleading as you used a high-resolution DEM instead of Lidar data. The source data for the DEM creation was Lidar, but not not essential for your study, therefore I recommend to change the title.

2. 1:17. The first sentence is in my opinion out of place and it would be better to state the objective of the study after introducing the general problem.

3. 2:8. Landsat8 → Landsat 8 (add space)

4. 2:18. It might be necessary to use the full name (Alaska) first and introduce the abbreviation. Adding the country name might be helpful for readers that are not familiar with US state abbreviations.

5. 2:24. You did not analyze Lidar imagery. It is a DEM derived from Lidar data.

6. 2:31. Before you used AK, here you write Alaska. Please try to be consistent or introduce the abbreviation at the first instance.

7. Section2 (2:30 ff). Fig S1: Can you provide a more detailed map (e.g. aerial/satellite image + bounding box) of the processed tile locations and probably some coordinates? Currently it is not possible to easily find your processed areas.

8. Section2 (2:30 ff). Could you provide more detail about the types of polygons? This information would be a good fit in this section. The Alaskan Arctic Coastal Plain Polygonal Geomorphology Map (Lara, 2015) + your own observations could be a good source for that.

9. 3:23. You use the term "trough". This term might work well for HCP, but LCPs also have rims. Using trough may not work well for the general variety of ice-wedge polygons and implies that you can only detect edges of HCP. Do the LCP still have small troughs between the rims? It seems so for at least some of the Polygons in your figures.

10. 3:24. "assigned a negative intensity proportional to its Euclidean distance from the closest trough". As this is a "distance transform" (to my knowledge) you could name it in parentheses. This would enhance the understanding of this part.

11. 4:2 Here you use both units (meters and pixels) in other cases you use only one of these. Please check if you could be somewhat more consistent.

12. 5:17 Double negation ("would not delineate any polygon whose center did not include") should be avoided.

13. 6:8 It would help if you could show you the location/extent of training data visually in your figures.

14. 6:12 "several iterations". Please be more specific.

15. 6:17. "we calculated the relative elevations of polygon centers at the Prudhoe Bay

training site". Why not both? I do not see the reason not doing it for Barrow.

16. 6:20 Section 4: The entire section is very vague and too qualitative. It completely lacks quantification of your results. Please add quantitative results and a proper accuracy assessment with independent training and validation areas to this section. The discussion is ok, but probably need some relation to similar studies and how your method performs in comparison with similar studies. Furthermore, it would be nice if you could discuss the transferability of your method to DEMs of other origin or spatial resolution.

17. 7:28. Here again, you are using DEM rather than Lidar

18. 7:30 "using a training workflow that can be completed in a single afternoon". One could argue if this sentence sounds quite sloppy. Maybe you could improve the style.

19. Please check the formal requirements if all sub-figures need to get enumerated instead of A/ and left/right

20. Figure 3: Do the colorized edges add any information? It clearly makes sense for polygons, but rather not for lines.

21. Figure 3: Legend/Colorbar: Adding the polygon type, LCP for negative, HCP for positive values (if I understand correctly) would help to understand Fig 3A more quickly.

22. Figure 3: "A" is hard to read with the colorful background. I suggest to either change the font color or add a box (or similar) in the background.

---

## Referee Comment (RC2) · W. Zhang (Referee) · 27 Sep 2018

Summary: The manuscript presents a machine learning-based workflow to perform extraction and measurement of ice wedge polygons (IWP) from digital elevation model (DEM). The major contribution of this study is the use of a state-of-the-art convolutional neural network (CNN) and other computer vision algorithms to map troughs and polygonal boundaries. A couple of major concerns must be addressed before considering for publication. Therefore, I suggest a major revision before further consideration.

1. Page 1-Line 22: troughs are not always polygonal boundaries. Polygonal boundaries can be rims.

2. Page 1-Line 23: "The same techniques could be applied to any form of remotely sensed data with sufficient spatial resolution". Please try not to make this kind of statement without any supportive evidence. For example, it's widely known that extracting actual boundaries of buildings in LiDAR data is much easier than even in very high resolution optical imagery.

3. Page 2-Line 11-19: As a technical paper with a focus on applications of machine learning/computer vision in Arctic IWP mapping, some related articles are not mentioned. Some methods have been previously proposed for mapping polygonal terrains, such as: (1) Pina P; Saraiva J.; Bandeira L.; Antunes J. (2008) "Polygonal terrains on Mars: A contribution to their geometric and topological characterization", Planetary and Space Science, 56, 1919-1924. (2) Bandeira L.; Pina P.; Saraiva J. (2010) "A multi-layer approach for the analysis of neighbourhood relations of polygons in remotely acquired images", Pattern Recognition Letters, 31, 1175-1183. (3) Zhang, W.; Witharana, C.; Liljedahl, A.; Kanevskiy, M. (2018). "Deep Convolutional Neural Networks for Automated Characterization of Arctic Ice-Wedge Polygons in Very High Spatial Resolution Aerial Imagery". Remote Sensing, 10(9), 1487.

4. Page 2-Line 11-19: The method (Mask R-CNN) used in the paper "Deep Convolutional Neural Networks for Automated Characterization of Arctic Ice-Wedge Polygons in Very High Spatial Resolution Aerial Imagery" is an end-to-end object instance segmentation mapping solution for optical RS imagery with much less required training data and steps than the proposed "Polygon delineation algorithm". Besides delineating IWPs precisely, the paper reports relatively accurate classification of IWP type. Please carefully justify the benefit of using the proposed workflow in the introduction section.

5. Page 2-Line 30: Please provide the size of two study areas in Study areas and data acquisition section instead of Page 6-Line 9.

6. Page 3-Line 20: It is difficult for readers to understand the workflow without a schematic. Please add a schematic of the proposed workflow in the Method section

because Figure 1 only shows some intermediate results.

7. Page 4-Line 1-5: Please explain each threshold setting here. For example, why did you set a radius as 20m in 2D filter; why did you set a minimum intensity to depression of 0.7 m or greater; . . .

8. Page 4-Line 9. "a CNN is a classification tool which accepts images of a fixed size (in our case, 27×27 grayscale arrays) as input and generates categorical 10 labels as output." From I see in Figure 1A, it does not look like a 27 rows by 27 columns input image.

9. Page 4-Section 3.1.2 Convolutional neural network. I suppose the authors conducted this study not earlier than 2017. (1) I wonder why the authors did not use an advanced CNN instead of a 7-layer CNN. A lot of advanced CNN architecture had been developed and shown amazing capacity in pattern recognition. Please provide reasons. (2) How many test data did you use to achieve 99% accuracy?

10. Figure 2. Edge misalignment is a common issue using any CNN to process a large picture. I do not see any edge misalignment issue in Figure 2. How did you stitch all partitioned (27*27 pixels) patches together?

11. Quantitative validation is necessary to assess the performance of the proposed workflow. Please provide an assessment of the delineation.

12. Figure 2: two study sites are two small extreme cases (1 km2) filled with IWPs. A case study with a diverse landscape and large area is expected to support the effectiveness of the proposed workflow. Please take your time and carefully address this concern.

13. Please make sure all maps have some essential components, such as scale bar and north arrow.

---

## Author Comment (AC1) · 2 Nov 2018

Dear Dr. Nitze,

Thank you very much for your review or our article and suggestions for improvement. To address several of your major concerns, we have entirely re-written the results and conclusion section of the manuscript, incorporating a quantitative evaluation of algorithm performance. At the request of the other reviewer, we have also extended the spatial extent of our analysis, and included a detailed comparison with a similarly focused study published in the last several weeks. Additionally, we have constructed a flow chart for our methods (Fig. 1) and provided the editor with a repository of our code and data, which we understand will be forwarded to you and the other referee. Please find attached our response (in bold) to the remainder of your comments (copied here in italics), followed by our revised manuscript and supplementary information.

**-Chuck Abolt**

General summary: The manuscript with the title "Rapid machine learning-based extraction and measurement of ice wedge polygons in airborne lidar data" describes the application of novel convolutional neural network (CNN) image recognition concepts for the delineation of ice-wedge polygons from DEM data. The methodology was tested in two different sites in northern Alaska. The application of state-of-the-art image recognition methods is a rather new and unexplored approach in remote sensing applications of the cryosphere. The paper has a strong technical focus and describes the methods thoroughly. Delineating ice-wedge polygon networks is an essential task for quantifying ground ice, predicting the resilience against degradation and understanding local scale hydrology. In contrast to the positive novelty of the paper, this paper lacks several major points. The authors remain very vague in the results sections, with practically no quantification or accuracy assessment of the results. To the reader it is hard to estimate the accuracy strengths and weaknesses of the method, as the results are presented rather qualitatively. Furthermore, the title suggests that the authors used Lidar data as their key input. This is somewhat misleading, as they used DEMs, which are based on Lidar data, but could be technically processed from other sources. As this manuscript has a technical focus I would really like to see a flowchart in this paper, as this will help to follow the processing chain much better. Furthermore, the authors did not mention any software (programming languages, packages) they used, which might be interesting for the readers. For the review process I am interested to see the code and the data. Overall I see a good potential for publication due to the interesting application of novel image recognition methods for delineating IW-polygons. However, the manuscript needs improvement in several sections, particularly in the results and discussion section. Therefore I recommend a major revision. Specific comments are stated below.

1. Title: The title is somewhat misleading as you used a high-resolution DEM instead of Lidar data. The source data for the DEM creation was Lidar, but not not essential for your study, therefore I recommend to change the title.

We have changed the title to "Rapid machine learning-based extraction and measurement of ice wedge polygons in high-resolution digital elevation models" (1:3). We have also changed the text in several spots to be more general, describing the source data as high-resolution DEMs rather than airborne lidar data.

2. 1:17. The first sentence is in my opinion out of place and it would be better to state the objective of the study after introducing the general problem.

**We have re-worded most of the first paragraph to focus generally on the problem before introducing our approach (1:20-2:3)**

3. 2:8. Landsat8→Landsat 8 (add space)

**Thank you. We have made the correction (2.20).**

4. 2:18. It might be necessary to use the full name (Alaska) first and introduce the abbreviation. Adding the country name might be helpful for readers that are not familiar with US state abbreviations.

**We now use the full name (Alaska) throughout the text.**

5. 2:24. You did not analyze Lidar imagery. It is a DEM derived from Lidar data.

**We have changed this wording throughout the text.**

6. 2:31. Before you used AK, here you write Alaska. Please try to be consistent or introduce the abbreviation at the first instance.

**We now use the full name (Alaska) throughout the text.**

7. Section2 (2:30 ff). Fig S1: Can you provide a more detailed map (e.g. aerial/satellite image + bounding box) of the processed tile locations and probably some coordinates? Currently it is not possible to easily find your processed areas.

At the current time we do not have permission to reveal the precise coordinates of the tiles from Prudhoe Bay. However, we have provided a more detailed map with bounding boxes of both airborne lidar surveys in Figure S1, which we hope will be useful for locating the general area of the analysis.

8. Section2 (2:30 ff). Could you provide more detail about the types of polygons? This information would be a good fit in this section. The Alaskan Arctic Coastal Plain Polygonal Geomorphology Map (Lara, 2015) + your own observations could be a good source for that.

We agree this information is useful and we have included descriptions of polygon geomorphology from both landscapes (4:1-2, 14-15). This section now includes a blend of our own observations and the classification performed by Lara et al. (2018).

9. 3:23. You use the term "trough". This term might work well for HCP, but LCPs also have rims. Using trough may not work well for the general variety of ice-wedge polygons and implies that you can only detect edges of HCP. Do the LCP still have small troughs between the rims? It seems so for at least some of the Polygons in your figures.

To improve clarity we have replaced references to "trough" and "non-trough" pixels throughout the text with the terms "boundary" and "non-boundary." We previously used the word "trough" almost synonymously with "polygonal boundary," as nearly all LCPs in our survey area are bound by shallow troughs, which are visible in the DEMs at 50 cm resolution.

10. 3:24. "assigned a negative intensity proportional to its Euclidean distance from the closest trough". As this is a "distance transform" (to my knowledge) you could name it in parentheses. This would enhance the understanding of this part.

Thank you. We now describe this procedure as a distance transform (4:29-30), (7:1), (7:31).

11. 4:2 Here you use both units (meters and pixels) in other cases you use only one of these. Please check if you could be somewhat more consistent.

To make our description more general, we now opt to use meters throughout the Methods section. The one exception is when we describe the size of the thumbnail images processed by the CNN, which we describe in both units for clarity (5:13-15).

12. 5:17 Double negation ("would not delineate any polygon whose center did not include") should be avoided.

We have rephrased this sentence to avoid the double negative (7:7-9).

13. 6:8 It would help if you could show you the location/extent of training data visually in your figures.

We now provide a more detailed map of the lidar survey areas in the supplement (Fig. S1).

14. 6:12 "several iterations". Please be more specific.

We now specify that it took four iterations to complete the bulk of the training at sites Barrow-1 and Prudhoe-1, then one more to "fine-tune" the CNN with additional examples of boundary and non-boundary features from each site (8:12-16), and another iteration to extend it across our remaining sites.

15. 6:17. "we calculated the relative elevations of polygon centers at the Prudhoe Bay training site". Why not both? I do not see the reason not doing it for Barrow.

**We now include calculations from both sites in Fig. 3.**

16. 6:20 Section 4: The entire section is very vague and too qualitative. It completely lacks quantification of your results. Please add quantitative results and a proper accuracy assessment with independent training and validation areas to this section. The discussion is ok, but probably need some relation to similar studies and how your method performs in comparison with similar studies. Furthermore, it would be nice if you could discuss the transferability of your method to DEMs of other origin or spatial resolution.

We have entirely re-written this section and included a manual validation procedure to quantitatively evaluate our results. We also extensively compare our CNN with a similarly focused study which was just published, and conduct an analysis comparing results using our approach to analyze DEMs at 25, 50, and 100 cm resolution (9:8 – 11:30).

17. 7:28. Here again, you are using DEM rather than Lidar

**We have corrected this throughout the text.**

18. 7:30 "using a training workflow that can be completed in a single afternoon". One could argue if this sentence sounds quite sloppy. Maybe you could improve the style.

**We have eliminated this phrase.**

19. Please check the formal requirements if all sub-figures need to get enumerated instead of A/ and left/right

We have now added letters designating subfigures to all four panels of Fig. 3.

20. Figure 3: Do the colorized edges add any information? It clearly makes sense for polygons, but rather not for lines.

We have gotten rid of the graphic showing colorized edges, and replaced the randomly colorized polygons in Fig. 3 in the main manuscript with polygons whose color designates relief.

21. Figure 3: Legend/Colorbar: Adding the polygon type, LCP for negative, HCP for positive values (if I understand correctly) would help to understand Fig 3A more quickly.

**We have added these labels to the color bar on the right side of Fig. 3.**

22. Figure 3: "A" is hard to read with the colorful background. I suggest to either change the font color or add a box (or similar) in the background.

We have condensed the previous Figs. 2 and 3 into one figure, and added white boxes to make the letters in Fig. 3 easier to read.

**Brief communication: Rapid machine learning-based extraction and measurement of ice wedge polygons in high-resolution digital elevation modelsairborne lidar data**

Charles J. Abolt1,2, Michael H. Young2, Adam LA. Atchley3, Cathy J. Wilson3

[revised manuscript text omitted]

---

## Author Comment (AC2) · 2 Nov 2018

Dear Dr. Zhang,

**Thank you very much for your review or our article. Please find below our responses (in bold) to your comments (in italics), followed by copies of the revised manuscript and supplemental information.**

**-Chuck Abolt**

*Summary: The manuscript presents a machine learning-based workflow to perform extraction and measurement of ice wedge polygons (IWP) from digital elevation model (DEM). The major contribution of this study is the use of a state-of-the-art convolutional neural network (CNN) and other computer vision algorithms to map troughs and polygonal boundaries. A couple of major concerns must be addressed before considering for publication. Therefore, I suggest a major revision before further consideration.*

*1. Page 1-Line 22: troughs are not always polygonal boundaries. Polygonal boundaries can be rims.*

**For clarity, we have replaced the terms "trough" and "not trough" with "boundary" and "non-boundary" throughout the text.**

*2. Page 1-Line 23: "The same techniques could be applied to any form of remotely sensed data with sufficient spatial resolution". Please try not to make this kind of statement without any supportive evidence. For example, it's widely known that extracting actual boundaries of buildings in LiDAR data is much easier than even in very high resolution optical imagery.*

**Thank you for this advice. We have removed this suggestion.**

*3. Page 2-Line 11-19: As a technical paper with a focus on applications of machine learning/computer vision in Arctic IWP mapping, some related articles are not mentioned. Some methods have been previously proposed for mapping polygonal terrains, such as: (1) Pina P; Saraiva J.; Bandeira L.; Antunes J. (2008) "Polygonal terrains on Mars: A contribution to their geometric and topological characterization", Planetary and Space Science, 56, 1919-1924. (2) Bandeira L.; Pina P.; Saraiva J. (2010) "A multi-layer approach for the analysis of neighbourhood relations of polygons in remotely acquired images", Pattern Recognition Letters, 31, 1175-1183. (3) Zhang, W.; Witharana, C.; Liljedahl, A.; Kanevskiy, M. (2018). "Deep Convolutional Neural Networks for Automated Characterization of Arctic Ice-Wedge Polygons in Very High Spatial Resolution Aerial Imagery". Remote Sensing, 10(9), 1487.*

**Thank you for suggesting these papers. As our manuscript is limited to 20 references, we did not have space to reference Bandeira et al. (2010), but we have included Pina et al. (2008) in our introduction (Page 2, Line 27). We also discuss**

Zhang et al. (2008) extensively in the introduction (Page 3, Lines 4-8), discussion (Section 4.4), and supplement (Text S1).

*4. Page 2-Line 11-19: The method (Mask R-CNN) used in the paper "Deep Convolutional Neural Networks for Automated Characterization of Arctic Ice-Wedge Polygons in Very High Spatial Resolution Aerial Imagery" is an end-to-end object instance segmentation mapping solution for optical RS imagery with much less required training data and steps than the proposed "Polygon delineation algorithm". Besides delineating IWPs precisely, the paper reports relatively accurate classification of IWP type. Please carefully justify the benefit of using the proposed workflow in the introduction section.*

Thank you for directing us to this paper. We were unaware of this study while we conducted our own, as it appears that it was published after we submitted our manuscript. We began applying CNNs to the problem of ice wedge polygon delineation in the spring of 2017, so we were also unaware at that time of Mask R-CNN, which seems to have been introduced later that year.

During the course of our study we briefly experimented with using more complex CNNs (such as fully convolutional networks). However, we opted to use a simpler architecture instead, as we found that it could deliver the results we sought when paired with the watershed segmentation technique pioneered by Pina et al. (2006). One of the main advantages of using the simpler tool was that training times were rapid (without employing transfer learning, we could train the CNN on 50 cm resolution data in less than 5 minutes on a personal laptop), which made it easy to improve skill iteratively by training the CNN, isolating areas where it performed poorly in a landscape, then incorporating new training data from those areas. We now emphasize this point in our abstract (Page 1, Lines 14-15) and introduction (Page 3, Lines 10-12), as well as in a flow chart of our methods (Fig. S1). Another advantage is that, because we use a watershed transform, our technique is amenable to mapping polygonal boundary networks, generating datasets for the same kinds of geometric analyses conducted by Pina et al. (2008) on Martial polygons.

We also believe that our method uses slightly less training data than Mask R-CNN. We elaborate on this point in the supplement (Text S1). In the discussion, we now dedicate a paragraph at the start of Section 4.4 to comparing our method with Mask R-CNN. While we believe that Mask R-CNN is better suited for pan-Arctic applications, we believe that our method is useful for generating unique measurements of microtopography in areas where high-resolution DEMs are available.

*5. Page 2-Line 30: Please provide the size of two study areas in Study areas and data acquisition section instead of Page 6-Line 9.*

We clarify now in the Study areas and data acquisition section that each study site is 1 $km^2$. We also include a figure in the supplement showing the bounding boxes for the lidar surveys from which the study areas were derived (Fig. S1).

*6. Page 3-Line 20: It is difficult for readers to understand the workflow without a schematic. Please add a schematic of the proposed workflow in the Method section because Figure 1 only shows some intermediate results.*

**We have added a chart summarizing our workflow to the manuscript (Fig. 1).**

*7. Page 4-Line 1-5: Please explain each threshold setting here. For example, why did you set a radius as 20m in 2D filter; why did you set a minimum intensity to depression of 0.7 m or greater; . . .*

**We have added justifications for each threshold setting applied in our Methods section.**

*8. Page 4-Line 9. "a CNN is a classification tool which accepts images of a fixed size (in our case, 27×27 grayscale arrays) as input and generates categorical 10 labels as output." From I see in Figure 1A, it does not look like a 27 rows by 27 columns input image.*

**We have augmented the Methods section and describe in more detail an intermediate step in our methods, in which a thumbnail image is created for every pixel in the de-trended elevation data, showing the 27 x 27 neighborhood around it. The CNN operates on these thumbnails and classifies each pixel as either boundary or not-boundary. Please see section 3.1.1 (Page 5, Lines 11-17). We have also reworded the caption of Fig. 1.**

*9. Page 4-Section 3.1.2 Convolutional neural network. I suppose the authors conducted this study not earlier than 2017. (1) I wonder why the authors did not use an advanced CNN instead of a 7-layer CNN. A lot of advanced CNN architecture had been developed and shown amazing capacity in pattern recognition. Please provide reasons. (2) How many test data did you use to achieve 99% accuracy?*

**Please note response to comment 4, and information on the training dataset in Text S1.**

*10. Figure 2. Edge misalignment is a common issue using any CNN to process a large picture. I do not see any edge misalignment issue in Figure 2. How did you stitch all partitioned (27*27 pixels) patches together?*

**Please see our answer above to comment 8. The 27 x 27 pixel images are not patches, but instead thumbnail images providing context for every pixel in the study area. After the CNN processes one thumbnail image for each pixel, it reassembles the pixel labels into a map of boundary and non-boundary pixels (Fig 1B).**

*11. Quantitative validation is necessary to assess the performance of the proposed workflow. Please provide an assessment of the delineation.*

**We have introduced a manual validation routine to assess accuracy (Page 8, Lines 24-32), and have rewritten the entire Results and discussion section to be more quantitative (Section 4).**

*12. Figure 2: two study sites are two small extreme cases (1 km2) filled with IWPs. A case study with a diverse landscape and large area is expected to support the effectiveness of the proposed workflow. Please take your time and carefully address this concern.*

**We have expanded the spatial extent of our analysis to 10 sites (two from Barrow and 8 from Prudhoe Bay) with diverse polygons and many non-polygonal landscape features. We show the results of delineation at all sites in the supplement (Figs. S2-S11). As described in the Discussion, our algorithm successfully partitions out features including thaw lakes, stream beds, non-polygonal marsh, shallow ponds, and the flood plain of a braided stream (Page 10, Lines 5-7).**

*13. Please make sure all maps have some essential components, such as scale bar and north arrow.*

**We have added a scale bar and North arrow to all DEMs (Figs. 3, S2-S11).**

[revised manuscript text omitted]

**Text S1.**  Comparison of training requirements and accuracy between CNN-watershed and Mask R-CNN algorithms.

Due to differences in the training and inference procedures used by each algorithm, training data requirements and accuracy are difficult to compare directly. Nonetheless, in several aspects, performance appears to be similar. In the present study, the CNN-watershed approach is trained initially on data derived from four manually-labeled 100 m tiles, representing 0.04 $km^2$. This training data is supplemented with extra examples of boundary and non-boundary features, the convex hulls of which sum to ~0.07 $km^2$, and the trained model is extrapolated across 10 $km^2$. The training to application ratio is therefore ~0.011, or 1.1%. In comparison, Mask R-CNN was trained on data from 340 90 m tiles, or ~2.75 $km^2$, then extrapolated across ~134 $km^2$, resulting in a training to application ratio of ~0.020 or 2.0% (Zhang et al., 2018). In general, within the area across which the CNN-watershed approach was applied, it was less likely than Mask R-CNN to fail to detect polygonal terrain, but more prone to mistakenly aggregate multiple ice wedge polygons into a single unit. These errors were particularly common at sites characterized by transitional terrain where ice wedge polygons grade into non-polygonal ground. It is reasonable to expect such mistakes in these areas, as microtopography is typically faint and polygons often appear to be bound incompletely by troughs. At one such site (Prudhoe-6), the number of incorrect conglomerate polygons by area delineated by the CNN-watershed algorithm was ~22% (Table 1). This number closely resembles the 21% of human-delineated polygons estimated to go undetected by Mask R-CNN in satellite-based optical imagery (Zhang et al., 2018).

**Table S1.**    Architecture of our CNN.

| Layer | Type | Neurons |
|---|---|---|
| 1 | Convolutional | 8 arrays of 27×27 |
| 2 | ReLU† | 8 arrays of 27×27 |
| 3 | Max-pooling | 8 arrays of 9×9 |
| 4 | ReLU | 8 arrays of 9×9 |
| 5 | Fully-connected | 64 |
| 6 | ReLU | 64 |
| 7 | Fully-connected | 2 |
| 8 | ReLU | 2 |
| 9 | Softmax | 2 |

† - ReLu – rectified linear unit

**Table 2.**  Results of manual validation at 100 cm and 25 cm resolution.

| Site | Polygons identified | Polygonal area (%) | % of polygons by instance | | | | % of polygons by area | | | |
| | | | Whole | Fractional | Conglomerate | Non-polygonal | Whole | Fractional | Conglomerate | Non-polygonal |
|---|---|---|---|---|---|---|---|---|---|---|
| Barrow-1 (100 cm) | 3058 | 74.3 | 73.4 | 18.6 | 6.2 | 1.8 | 65.5 | 16.6 | 13.4 | 4.1 |
| Prudhoe-1 (100 cm) | 3019 | 100 | 85.6 | 11.8 | 2.6 | 0.0 | 88.0 | 79.7 | 4.0 | 0.0 |
| Barrow-1 (25 cm) | 2870 | 71.6 | 89.0 | 3.8 | 3.4 | 3.8 | 83.4 | 2.0 | 9.7 | 4.8 |
| Prudhoe-1 (25 cm) | 3193 | 100 | 93.6 | 3.4 | 2.4 | 0.6 | 94.0 | 1.6 | 4.3 | 0.1 |

---

## Referee Comment (RC3) · I. Nitze (Referee) · 12 Nov 2018

The authors did a good job addressing the reviewers' comments and concerns. They rewrote large parts of the manuscript to significantly improve the manuscript, especially the validation part. In its current state, the manuscript only needs some minor editorial improvements and clarifications. In some places the wording/style may need some improvement. Therefore I recommend a minor revision. Ingmar Nitze

Specific comments:

Page and line numbers (P:L) refer to the latest manuscript version with tracked

changes.

3:4: I think it would be good to have one or two more (recent) examples of CNNs. This would strengthen your point using CNNs.

6:8: It would be more consistent if you use rather "100 x 100" m instead of "100m" edges. For your thumbnails and filter sizes you also use "n x n".

7:6: neighbour (British English, you used American English otherwise)

7:8: "tend measure" –> "tend to measure"

10:18: Somewhat…somewhat. This sentence may need some slight style improvement.

10:22: I don't really like the word "crisp", maybe use some better term, which describes that 25cm is a sufficient resolution for your target.

12:2: "A relatively simple CNN is capable …". I think here you should also mention that you applied some image processing techniques, as they are also important for you workflow in my opinion.

Figure 3 B and D: The color scale now looks very nice, but be aware of color-blindness (probably avoid red to green).

Table 1: It would be great to quickly indicate the size of each site in the caption. This would help to put the # of polygons into context and to interpret the table without the text.

Figure S1: I would say it is more commonplace to use "Easting" and "Northing" without "s". I think geographical coordinates (Latitude and Longitude) would be nicer and easier to find for most readers. A scalebar would be helpful. An inset box showing the site's location within Alaska would also help.

---

## Referee Comment (RC4) · W. Zhang (Referee) · 15 Nov 2018

Summary: This paper entitled "Brief communication: Rapid machine learning-based extraction and measurement of ice wedge polygons in airborne lidar data" presents a workflow for rapid delineation and microtopographic characterization of ice wedge polygons within high-resolution digital elevation models. The authors have been addressed all my concerns in their revised version. The paper is well conducted and presented. In summary, I think this paper is publishable in The Cryosphere. Please consider improving two places before final publishing.

Minor comments: 1. Title: Is it necessary to include "brief communication" in the title?

[Figure]

The authors creatively proposed a new way to rapidly delineate and microtopographic characterize ice wedge polygons rather than summarizing previous work. Please remove "brief communication" if possible. 2. Abstract: Please consider including some validation result with numbers in the abstract.

---

## Author Comment (AC3) · 10 Dec 2018

The authors did a good job addressing the reviewers' comments and concerns. They rewrote large parts of the manuscript to significantly improve the manuscript, especially the validation part. In its current state, the manuscript only needs some minor editorial improvements and clarifications. In some places the wording/style may need some improvement. Therefore I recommend a minor revision. Ingmar Nitze

Specific comments: Page and line numbers (P:L) refer to the latest manuscript version with tracked changes.

3:4: I think it would be good to have one or two more (recent) examples of CNNs. This would strengthen your point using CNNs.

**We have inserted two references from 2018 which apply CNNs to extract road networks from satellite imagery, a similar task to extracting polygon boundaries from lidar DEMs (3:4-5).**

6:8: It would be more consistent if you use rather "100 x 100" m instead of "100m" edges. For your thumbnails and filter sizes you also use "n x n".

**We have changed this wording throughout the manuscript (6:9, 6:17, 8:11, Text S1).**

7:6: neighbour (British English, you used American English otherwise)

**Thank you. We have changed to "neighbor" (7:7).**

7:8: "tend measure" –> "tend to measure"

**Thank you. This has been fixed (7:9).**

10:18: Somewhat . . . somewhat. This sentence may need some slight style improvement.

**We have reworded this sentence to only use the word "somewhat" once (10:19-20).**

10:22: I don't really like the word "crisp", maybe use some better term, which describes that 25cm is a sufficient resolution for your target.

**We have eliminated the word "crisp," instead stating that the imagery "is of sufficient resolution for our purposes" (10:23).**

12:2: "A relatively simple CNN is capable . . .". I think here you should also mention that you applied some image processing techniques, as they are also important for you workflow in my opinion.

**We now begin this sentence saying "A relatively simple CNN paired with a set of common image processing techniques…" (13:3).**

Figure 3 B and D: The color scale now looks very nice, but be aware of color-blindness (probably avoid red to green).

**Thank you for this suggestion. Fig. 3 now uses a red to blue color scale.**

Table 1: It would be great to quickly indicate the size of each site in the caption. This would help to put the # of polygons into context and to interpret the table without the text.

**We have inserted this information into the captions of Table 1 and Table S2.**

Figure S1: I would say it is more commonplace to use "Easting" and "Northing" without "s". I think geographical coordinates (Latitude and Longitude) would be nicer and easier to find for most readers. A scalebar would be helpful. An inset box showing the site's location within Alaska would also help.

**Thank you for these suggestions. Fig. S1 now uses geographic coordinates and includes and inset map and a scale bar. The words "Easting" and "Northing" have been eliminated.**

[revised manuscript text omitted]

**A**
**B**

| 79 | elevation (m a.s.l.) | 83 |

| -0.7 | relief (m) | 0.7 |

**Figure S12.** 50 cm DEM of the Prudhoe Bay training site before (**a**) and after (**b**) removing regional trends to isolate microtopography.

[Figure]

**Figure S13.** Samples of manually delineated data used to train the CNN, including a tile in which troughs are fully delineated (**a**) and a tile used to supplement the training deck with extra examples of non-trough pixels (**b**).

[Figure]

**Figure S14.** Delineation algorithm on the same ice wedge polygon at 100 cm (**a**), 50 cm (**b**), and 25 cm (**c**) resolution. Each image is 40 m across. Note that anomalously low pixels in the polygon center in (**a**) are mistaken as polygon boundaries, incorrectly fragmenting the polygon.

**Text S1.**     Comparison of training requirements and accuracy between CNN-watershed and Mask R-CNN algorithms.

Due to differences in the training and inference procedures used by each algorithm, training data requirements and accuracy are difficult to compare directly. Nonetheless, in several aspects, performance appears to be similar. In the present study, the CNN-watershed approach is trained initially on data derived from four manually-labeled 100 × 100 m tiles, representing 0.04 km$^2$. This training data is supplemented with extra examples of boundary and non-boundary features, the convex hulls of which sum to ~0.07 km$^2$, and the trained model is extrapolated across 10 km$^2$. The training to application ratio is therefore ~0.011, or 1.1%. In comparison, Mask R-CNN was trained on data from 340 90 × 90 m tiles, or ~2.75 km$^2$, then extrapolated across ~134 km$^2$, resulting in a training to application ratio of ~0.020 or 2.0% (Zhang et al., 2018). In general, within the area across which the CNN-watershed approach was applied, it was less likely than Mask R-CNN to fail to detect polygonal terrain, but more prone to mistakenly aggregate multiple ice wedge polygons into a single unit. These errors were particularly common at sites characterized by transitional terrain where ice wedge polygons grade into non-polygonal ground. It is reasonable to expect such mistakes in these areas, as microtopography is typically faint and polygons often appear to be bound incompletely by troughs. At one such site (Prudhoe-6), the number of incorrect conglomerate polygons by area delineated by the CNN-watershed algorithm was ~22% (Table 1). This number closely resembles the 21% of human-delineated polygons estimated to go undetected by Mask R-CNN in satellite-based optical imagery (Zhang et al., 2018).

**Table S1.**     Architecture of our CNN.

| Layer | Type | Neurons |
|---|---|---|
| 1 | Convolutional | 8 arrays of 27×27 |
| 2 | ReLU† | 8 arrays of 27×27 |
| 3 | Max-pooling | 8 arrays of 9×9 |
| 4 | ReLU | 8 arrays of 9×9 |
| 5 | Fully-connected | 64 |
| 6 | ReLU | 64 |
| 7 | Fully-connected | 2 |
| 8 | ReLU | 2 |
| 9 | Softmax | 2 |

† - ReLu – rectified linear unit

**Table S2.**    Results of manual validation at 100 cm and 25 cm resolution (sites are 1 km$^2$).

| Site | Polygons identified | Polygonal area (%) | % of polygons by instance | | | | % of polygons by area | | | |
|---|---|---|---|---|---|---|---|---|---|---|
| | | | Whole | Fractional | Conglomerate | Non-polygonal | Whole | Fractional | Conglomerate | Non-polygonal |
| Barrow-1 (100 cm) | 3058 | 74.3 | 73.4 | 18.6 | 6.2 | 1.8 | 65.5 | 16.6 | 13.4 | 4.1 |
| Prudhoe-1 (100 cm) | 3019 | 100 | 85.6 | 11.8 | 2.6 | 0.0 | 88.0 | 79.7 | 4.0 | 0.0 |
| Barrow-1 (25 cm) | 2870 | 71.6 | 89.0 | 3.8 | 3.4 | 3.8 | 83.4 | 2.0 | 9.7 | 4.8 |
| Prudhoe-1 (25 cm) | 3193 | 100 | 93.6 | 3.4 | 2.4 | 0.6 | 94.0 | 1.6 | 4.3 | 0.1 |

[Figure]

**Figure S1.** Bounding boxes of airborne lidar surveys .

[Figure]

**Figure S2.**   50 cm DEM (**a**) and polygon delineation (**b**) at site Barrow-1.

[Figure]

**Figure S3.** 50 cm DEM (**a**) and polygon delineation (**b**) at site Barrow-2.

[Figure]

**Figure S4.**    50 cm DEM (**a**) and polygon delineation (**b**) at site Prudhoe-1.

[Figure]

**Figure S5.**     50 cm DEM (**a**) and polygon delineation (**b**) at site Prudhoe-2.

[Figure]

**Figure S6.** 50 cm DEM (**a**) and polygon delineation (**b**) at site Prudhoe-3.

[Figure]

**Figure S7.** 50 cm DEM (**a**) and polygon delineation (**b**) at site Prudhoe-4.

[Figure]

**Figure S8.**    50 cm DEM (**a**) and polygon delineation (**b**) at site Prudhoe-5.

[Figure]

**Figure S9.** 50 cm DEM (**a**) and polygon delineation (**b**) at site Prudhoe-6.

[Figure]

**Figure S10.** 50 cm DEM (**a**) and polygon delineation (**b**) at site Prudhoe-7.

[Figure]

**Figure S11.** 50 cm DEM (**a**) and polygon delineation (**b**) at site Prudhoe-8.

[Figure]

79    elevation (m a.s.l.)    83        -0.7    relief (m)    0.7

**Figure S12.** 50 cm DEM of the Prudhoe Bay training site before (**a**) and after (**b**) removing regional trends to isolate microtopography.

[Figure]

**Figure S13.** Samples of manually delineated data used to train the CNN, including a tile in which troughs are fully delineated (**a**) and a tile used to supplement the training deck with extra examples of non-trough pixels (**b**).

[Figure]

**Figure S14.** Delineation algorithm on the same ice wedge polygon at 100 cm (**a**), 50 cm (**b**), and 25 cm (**c**) resolution. Each image is 40 m across. Note that anomalously low pixels in the polygon center in (**a**) are mistaken as polygon boundaries, incorrectly fragmenting the polygon.

**Text S1.**     Comparison of training requirements and accuracy between CNN-watershed and Mask R-CNN algorithms.

        Due to differences in the training and inference procedures used by each algorithm, training data requirements and accuracy are difficult to compare directly. Nonetheless, in several aspects, performance appears to be similar. In the present study, the CNN-watershed approach is trained initially on data derived from four manually-labeled 100 × 100 m tiles, representing 0.04 km$^2$. This training data is supplemented with extra examples of boundary and non-boundary features, the convex hulls of which sum to ~0.07 km$^2$, and the trained model is extrapolated across 10 km$^2$. The training to application ratio is therefore ~0.011, or 1.1%. In comparison, Mask R-CNN was trained on data from 340 90 × 90 m tiles, or ~2.75 km$^2$, then extrapolated across ~134 km$^2$, resulting in a training to application ratio of ~0.020 or 2.0% (Zhang et al., 2018). In general, within the area across which the CNN-watershed approach was applied, it was less likely than Mask R-CNN to fail to detect polygonal terrain, but more prone to mistakenly aggregate multiple ice wedge polygons into a single unit. These errors were particularly common at sites characterized by transitional terrain where ice wedge polygons grade into non-polygonal ground. It is reasonable to expect such mistakes in these areas, as microtopography is typically faint and polygons often appear to be bound incompletely by troughs. At one such site (Prudhoe-6), the number of incorrect conglomerate polygons by area delineated by the CNN-watershed algorithm was ~22% (Table 1). This number closely resembles the 21% of human-delineated polygons estimated to go undetected by Mask R-CNN in satellite-based optical imagery (Zhang et al., 2018).

**Table S1.**     Architecture of our CNN.

| Layer | Type | Neurons |
|---|---|---|
| 1 | Convolutional | 8 arrays of 27×27 |
| 2 | ReLU† | 8 arrays of 27×27 |
| 3 | Max-pooling | 8 arrays of 9×9 |
| 4 | ReLU | 8 arrays of 9×9 |
| 5 | Fully-connected | 64 |
| 6 | ReLU | 64 |
| 7 | Fully-connected | 2 |
| 8 | ReLU | 2 |
| 9 | Softmax | 2 |

† - ReLu – rectified linear unit

**Table S2.**    Results of manual validation at 100 cm and 25 cm resolution (sites are 1 km²).

| Site | Polygons identified | Polygonal area (%) | % of polygons by instance | | | | % of polygons by area | | | |
| | | | Whole | Fractional | Conglomerate | Non-polygonal | Whole | Fractional | Conglomerate | Non-polygonal |
|---|---|---|---|---|---|---|---|---|---|---|
| Barrow-1 (100 cm) | 3058 | 74.3 | 73.4 | 18.6 | 6.2 | 1.8 | 65.5 | 16.6 | 13.4 | 4.1 |
| Prudhoe-1 (100 cm) | 3019 | 100 | 85.6 | 11.8 | 2.6 | 0.0 | 88.0 | 79.7 | 4.0 | 0.0 |
| Barrow-1 (25 cm) | 2870 | 71.6 | 89.0 | 3.8 | 3.4 | 3.8 | 83.4 | 2.0 | 9.7 | 4.8 |
| Prudhoe-1 (25 cm) | 3193 | 100 | 93.6 | 3.4 | 2.4 | 0.6 | 94.0 | 1.6 | 4.3 | 0.1 |

---

## Author Comment (AC4) · 10 Dec 2018

Summary: This paper entitled "Brief communication: Rapid machine learning-based extraction and measurement of ice wedge polygons in airborne lidar data" presents a workflow for rapid delineation and microtopographic characterization of ice wedge polygons within high-resolution digital elevation models. The authors have been addressed all my concerns in their revised version. The paper is well conducted and presented. In summary, I think this paper is publishable in The Cryosphere. Please consider improving two places before final publishing.

Minor comments:
1. Title: Is it necessary to include "brief communication" in the title? The authors creatively proposed a new way to rapidly delineate and microtopographic characterize ice wedge polygons rather than summarizing previous work. Please remove "brief communication" if possible.

**We originally submitted the manuscript as a "brief communication" as it was somewhat shorter than most regular manuscripts. We are comfortable with either changing the status of the manuscript or keeping it a brief communication, depending on the opinion of the editor.**

2. Abstract: Please consider including some validation result with numbers in the abstract.

**We now include a statement in the abstract that manual validations indicate 70-96% accuracy by area at the kilometer scale.**

[revised manuscript text omitted]

**Text S1.**      Comparison of training requirements and accuracy between CNN-watershed and Mask R-CNN algorithms.

Due to differences in the training and inference procedures used by each algorithm, training data requirements and accuracy are difficult to compare directly. Nonetheless, in several aspects, performance appears to be similar. In the present study, the CNN-watershed approach is trained initially on data derived from four manually-labeled 100 × 100 m tiles, representing 0.04 km$^2$. This training data is supplemented with extra examples of boundary and non-boundary features, the convex hulls of which sum to ~0.07 km$^2$, and the trained model is extrapolated across 10 km$^2$. The training to application ratio is therefore ~0.011, or 1.1%. In comparison, Mask R-CNN was trained on data from 340 90 × 90 m tiles, or ~2.75 km$^2$, then extrapolated across ~134 km$^2$, resulting in a training to application ratio of ~0.020 or 2.0% (Zhang et al., 2018). In general, within the area across which the CNN-watershed approach was applied, it was less likely than Mask R-CNN to fail to detect polygonal terrain, but more prone to mistakenly aggregate multiple ice wedge polygons into a single unit. These errors were particularly common at sites characterized by transitional terrain where ice wedge polygons grade into non-polygonal ground. It is reasonable to expect such mistakes in these areas, as microtopography is typically faint and polygons often appear to be bound incompletely by troughs. At one such site (Prudhoe-6), the number of incorrect conglomerate polygons by area delineated by the CNN-watershed algorithm was ~22% (Table 1). This number closely resembles the 21% of human-delineated polygons estimated to go undetected by Mask R-CNN in satellite-based optical imagery (Zhang et al., 2018).

**Table S1.**      Architecture of our CNN.

| Layer | Type | Neurons |
|-------|------|---------|
| 1 | Convolutional | 8 arrays of 27×27 |
| 2 | ReLU† | 8 arrays of 27×27 |
| 3 | Max-pooling | 8 arrays of 9×9 |
| 4 | ReLU | 8 arrays of 9×9 |
| 5 | Fully-connected | 64 |
| 6 | ReLU | 64 |
| 7 | Fully-connected | 2 |
| 8 | ReLU | 2 |
| 9 | Softmax | 2 |

† - ReLu – rectified linear unit

**Table S2.**  Results of manual validation at 100 cm and 25 cm resolution (sites are 1 km$^2$).

| Site | Polygons identified | Polygonal area (%) | % of polygons by instance | | | | % of polygons by area | | | |
|------|---------------------|--------------------|---------|------------|--------------|---------------|-------|------------|--------------|---------------|
| | | | Whole | Fractional | Conglomerate | Non-polygonal | Whole | Fractional | Conglomerate | Non-polygonal |
| Barrow-1 (100 cm) | 3058 | 74.3 | 73.4 | 18.6 | 6.2 | 1.8 | 65.5 | 16.6 | 13.4 | 4.1 |
| Prudhoe-1 (100 cm) | 3019 | 100 | 85.6 | 11.8 | 2.6 | 0.0 | 88.0 | 79.7 | 4.0 | 0.0 |
| Barrow-1 (25 cm) | 2870 | 71.6 | 89.0 | 3.8 | 3.4 | 3.8 | 83.4 | 2.0 | 9.7 | 4.8 |
| Prudhoe-1 (25 cm) | 3193 | 100 | 93.6 | 3.4 | 2.4 | 0.6 | 94.0 | 1.6 | 4.3 | 0.1 |

[Figure]

**Figure S1.**    Bounding boxes of airborne lidar surveys .

[Figure]

**Figure S2.** 50 cm DEM (**a**) and polygon delineation (**b**) at site Barrow-1.

[Figure]

**Figure S3.**    50 cm DEM (**a**) and polygon delineation (**b**) at site Barrow-2.

[Figure]

**Figure S4.** 50 cm DEM (**a**) and polygon delineation (**b**) at site Prudhoe-1.

[Figure]

**Figure S5.** 50 cm DEM (**a**) and polygon delineation (**b**) at site Prudhoe-2.

[Figure]

**Figure S6.**     50 cm DEM (**a**) and polygon delineation (**b**) at site Prudhoe-3.

[Figure]

**Figure S7.** 50 cm DEM (**a**) and polygon delineation (**b**) at site Prudhoe-4.

[Figure]

**Figure S8.**     50 cm DEM (**a**) and polygon delineation (**b**) at site Prudhoe-5.

[Figure]

**Figure S9.** 50 cm DEM (**a**) and polygon delineation (**b**) at site Prudhoe-6.

[Figure]

**Figure S10.**  50 cm DEM (**a**) and polygon delineation (**b**) at site Prudhoe-7.

[Figure]

**Figure S11.** 50 cm DEM (**a**) and polygon delineation (**b**) at site Prudhoe-8.

[Figure]

**Figure S12.** 50 cm DEM of the Prudhoe Bay training site before (**a**) and after (**b**) removing regional trends to isolate microtopography.

[Figure]

**Figure S13.** Samples of manually delineated data used to train the CNN, including a tile in which troughs are fully delineated (**a**) and a tile used to supplement the training deck with extra examples of non-trough pixels (**b**).

[Figure]

**Figure S14.** Delineation algorithm on the same ice wedge polygon at 100 cm (**a**), 50 cm (**b**), and 25 cm (**c**) resolution. Each image is 40 m across. Note that anomalously low pixels in the polygon center in (**a**) are mistaken as polygon boundaries, incorrectly fragmenting the polygon.

**Text S1.**     Comparison of training requirements and accuracy between CNN-watershed and Mask R-CNN algorithms.

Due to differences in the training and inference procedures used by each algorithm, training data requirements and accuracy are difficult to compare directly. Nonetheless, in several aspects, performance appears to be similar. In the present study, the CNN-watershed approach is trained initially on data derived from four manually-labeled 100 × 100 m tiles, representing 0.04 km$^2$. This training data is supplemented with extra examples of boundary and non-boundary features, the convex hulls of which sum to ~0.07 km$^2$, and the trained model is extrapolated across 10 km$^2$. The training to application ratio is therefore ~0.011, or 1.1%. In comparison, Mask R-CNN was trained on data from 340 90 × 90 m tiles, or ~2.75 km$^2$, then extrapolated across ~134 km$^2$, resulting in a training to application ratio of ~0.020 or 2.0% (Zhang et al., 2018). In general, within the area across which the CNN-watershed approach was applied, it was less likely than Mask R-CNN to fail to detect polygonal terrain, but more prone to mistakenly aggregate multiple ice wedge polygons into a single unit. These errors were particularly common at sites characterized by transitional terrain where ice wedge polygons grade into non-polygonal ground. It is reasonable to expect such mistakes in these areas, as microtopography is typically faint and polygons often appear to be bound incompletely by troughs. At one such site (Prudhoe-6), the number of incorrect conglomerate polygons by area delineated by the CNN-watershed algorithm was ~22% (Table 1). This number closely resembles the 21% of human-delineated polygons estimated to go undetected by Mask R-CNN in satellite-based optical imagery (Zhang et al., 2018).

**Table S1.**    Architecture of our CNN.

| Layer | Type | Neurons |
|---|---|---|
| 1 | Convolutional | 8 arrays of 27×27 |
| 2 | ReLU† | 8 arrays of 27×27 |
| 3 | Max-pooling | 8 arrays of 9×9 |
| 4 | ReLU | 8 arrays of 9×9 |
| 5 | Fully-connected | 64 |
| 6 | ReLU | 64 |
| 7 | Fully-connected | 2 |
| 8 | ReLU | 2 |
| 9 | Softmax | 2 |

† - ReLu – rectified linear unit

**Table S2.**     Results of manual validation at 100 cm and 25 cm resolution (sites are 1 km$^2$).

| Site | Polygons identified | Polygonal area (%) | % of polygons by instance | | | | % of polygons by area | | | |
|---|---|---|---|---|---|---|---|---|---|---|
| | | | Whole | Fractional | Conglomerate | Non-polygonal | Whole | Fractional | Conglomerate | Non-polygonal |
| Barrow-1 (100 cm) | 3058 | 74.3 | 73.4 | 18.6 | 6.2 | 1.8 | 65.5 | 16.6 | 13.4 | 4.1 |
| Prudhoe-1 (100 cm) | 3019 | 100 | 85.6 | 11.8 | 2.6 | 0.0 | 88.0 | 79.7 | 4.0 | 0.0 |
| Barrow-1 (25 cm) | 2870 | 71.6 | 89.0 | 3.8 | 3.4 | 3.8 | 83.4 | 2.0 | 9.7 | 4.8 |
| Prudhoe-1 (25 cm) | 3193 | 100 | 93.6 | 3.4 | 2.4 | 0.6 | 94.0 | 1.6 | 4.3 | 0.1 |